# New Insights into Microglial Mechanisms of Memory Impairment in Alzheimer’s Disease

**DOI:** 10.3390/biom12111722

**Published:** 2022-11-21

**Authors:** Na Li, Mingru Deng, Gonghui Hu, Nan Li, Haicheng Yuan, Yu Zhou

**Affiliations:** 1Department of Rehabilitation Medicine, Affiliated Hospital of Qingdao University, Qingdao 266000, China; 2Department of Medicine, Qingdao Binhai University, Qingdao 266555, China; 3Department of Physiology and Pathophysiology, School of Basic Medical Sciences, Qingdao University, Qingdao 266071, China; 4Department of Neurology, Affiliated Qingdao Central Hospital of Qingdao University, Qingdao 266042, China; 5Department of Health and Life Sciences, University of Health and Rehabilitation Sciences, Qingdao 266000, China; 6Institute of Brain Sciences and Related Disorders, Qingdao University, Qingdao 266071, China

**Keywords:** Alzheimer’s disease, microglia, memory impairment, inflammation, reviews

## Abstract

Alzheimer’s disease (AD) is the most common progressive and irreversible neurodegeneration characterized by the impairment of memory and cognition. Despite years of studies, no effective treatment and prevention strategies are available yet. Identifying new AD therapeutic targets is crucial for better elucidating the pathogenesis and establishing a valid treatment of AD. Growing evidence suggests that microglia play a critical role in AD. Microglia are resident macrophages in the central nervous system (CNS), and their core properties supporting main biological functions include surveillance, phagocytosis, and the release of soluble factors. Activated microglia not only directly mediate the central immune response, but also participate in the pathological changes of AD, including amyloid-beta (Aβ) aggregation, tau protein phosphorylation, synaptic dissection, neuron loss, memory function decline, etc. Based on these recent findings, we provide a new framework to summarize the role of microglia in AD memory impairment. This evidence suggests that microglia have the potential to become new targets for AD therapy.

## 1. Introduction

Alzheimer’s disease (AD) is the most common neurodegenerative disorder worldwide closely related to age, and its main clinical features are irreversible memory loss, cognitive dysfunction, and personality changes [1]. AD mostly occurs in elderly individuals over 65 years old and seriously affects their ability in daily life. AD is now a major medical and social problem worldwide [2]. The main pathological features of AD are the deposition of amyloid beta (Aβ) in the brain and the formation of neurofibrillary tangles (NFTs) formed by hyperphosphorylated tau protein [3]. Aβ and NFT can cause oxidative stress, neuro-inflammation, mitochondrial dysfunction, and an imbalance of biometal homeostasis [4,5,6], ultimately leading to synaptic loss, dendritic spine reduction, and neuronal death [7]. Despite extensive research over the years, the underlying pathogenesis of AD remains unclear. The current treatment for AD still targets Aβ, but clinical trials have failed [8]. Once AD is initiated, not a single drug can stop or reverse disease progression, meaning that new molecular targets urgently need to be identified. The study of novel and highly effective molecular targets for AD has become a global focus.

Since Alois Alzheimer first discovered that microglia are associated with Aβ plaques in the brains of AD patients [9], microglia have attracted much attention. Wegiel found that the number of microglia in the brains of patients with AD was positively correlated with the size of Aβ plaques [10] and the activated microglia clustered around them [11]. All these findings indicate that microglial activation is a key feature of AD. Studies using positron emission tomography (PET) techniques to diagnose AD found that activation of microglia in the brains of patients with AD precedes cognitive and memory decline [12], indicating that microglia may be involved in the pathogenesis of the disease. Increasing recent evidence shows that microglia are closely related to AD memory impairment.

A close spatiotemporal relationship between Aβ and microglia was found in both AD animal models and AD patients [13,14,15]. Microglia tightly wrap around Aβ plaques to form a barrier, preventing outward development, and thus, limiting the toxic effects of the plaques on peripheral neurons [16]. Microglia-induced neuro-inflammation directly damages neurons while promoting abnormal protein aggregation in AD patient brains [17]. Aβ was found in microglial cells, providing direct evidence that microglia clear Aβ by phagocytosis [18]. Genome-wide association studies (GWAS) found many AD-related single-nucleotide polymorphisms (SNPs) associated with microglia, among which 29 have been identified as highly expressed in microglia [19], such as triggering receptor expressed on myeloid cells 2 (TREM2), myeloid cell surface antigen CD33, and ATP-binding cassette subfamily A Member 7 (ABCA7) [20,21]. All these findings indicate that microglia may play a key role in the occurrence and development of AD. Therefore, microglia have the potential to become a new target for AD treatment. In this review, we begin by illustrating microglia’s origin, morphology states, and physiological functions, with an emphasis on their functions in the adult brain; then, we conduct a detailed discussion about the presently proposed mechanism of microglial contribution to AD memory deficits. By summarizing the recent findings about the causal relationship between microglia and AD, and the related clinical implications, we hope that this study can strengthen the understanding on the important role of microglia in AD-related synaptic dysfunction and cognitive decline. As the clinical trials for drugs targeting Aβ have all failed, it is plausible to rethink the potential of microglia manipulation as a therapeutic to ameliorate the cognitive deficits associated with AD.

## 2. Origin and Physiological Function of Microglia

### 2.1. Origin and Morphology of Microglia

Microglia are distributed throughout the central nervous system (CNS), accounting for approximately 5–12% of the total number of brain cells [9]. As shown in Figure 1, mammalian microglia originate from primitive yolk sac progenitor cells [22]. During early embryonic development, microglia start invading the neuroepithelium at E8.5 in mice [23]. Microglia demonstrate an ameboid morphology during fetal brain development, which is important for their mobility and the phagocytosis of debris resulting from apoptosis and synaptic pruning. During the perinatal stage, microglia migrate throughout the CNS to form a regular network that spreads across the three-dimensional space of the mouse brain parenchyma. By postnatal day 28 (P28), microglia mature and transition into a highly ramified morphology characterized by a small cell body and multiple long, thin-branched processes that constantly protrude and retract from the soma. Microglia maintain this highly ramified morphology throughout life to survey the entire brain. Once encountering stimulation, activated microglia can transform back into the ameboid-like morphology characterized by a large, rounded cell body with fewer, thicker, and shorter processes, similar to that observed early in brain development [24]. Although microglial morphology does not equate with microglial function, such as phagocytosis, the change in morphology provides an important sign of function alteration.

Microglia have the capacity to maintain population numbers by local self-renewal, a proliferation process independent of recruitment, allowing them to repopulate the CNS within 1 week of depletion [25]. Expression of the colony-stimulating factor 1 receptor (CSF1R) is critical for microglial development and maintenance. Two cytokines, CSF1 and interleukin-34 (IL-34), are ligands of CSF1R. IL-34 is produced by neurons, while CSF1 is primarily secreted by oligodendrocytes and astrocytes. Intriguingly, matured microglia in different brain regions exhibit regional distinction in density, function, and molecular markers [25,26]. The physiological significance of this spatial difference remains uncertain [27,28].

In summary, mammalian microglia are yolk-sac-derived, long-lived cells within the CNS parenchyma that persist into adulthood and self-renew at a steady state. The identification of microglia is currently based on highly enriched genes expressed in microglia, also called “microglial markers”, which include transcription factors Pu.1; cytoplasmic markers such as ionized calcium-binding adapter molecule 1 (IBA1); and surface markers such as the purinergic receptor P2YR12, transmembrane protein 119 (TMEM119), and CSF1R [23].

### 2.2. Multidimensional States and Phenotypes of Microglia

Microglia were once described as a physiological (“resting”) state versus a pathological (“activated”) state [29]. However, it is now realized that microglia are never resting. Instead, they are the most dynamic cells in the healthy mature brain, continuously surveying the parenchyma with their highly motile processes, and exceptionally responsive to alterations in their local environment, even in the absence of pathological challenge [30].

Traditionally, microglial activation was classified into M1 and M2 states. M1 activation is induced by interferon-γ (IFN-γ) and lipopolysaccharide (LPS). Microglia in this state mainly produce inflammatory cytokines and chemokines, such as tumor necrosis factor alpha (TNF-α), interleukin (IL)-6, IL-1β, IL-12, and CC chemokine ligand 2 (CCL-2). They also express nicotinamide adenine dinucleotide phosphate (NADPH) oxidase and inducible nitric oxide synthase (iNOS); major histocompatibility complex-II (MHC-II); integrins (CD11b, CD11c); co-stimulatory molecules (CD36, CD45, CD47); and Fc receptors, while M2 activation is induced by anti-inflammatory cytokines such as IL-4 and IL-13. Microglia in this state mainly produce anti-inflammatory cytokines IL-10 and transforming growth factor (TGF)-β. They also produce growth factors such as insulin-like growth factor-1 (IGF-1), fibroblast growth factor (FGF), CSF1R, and neurotrophic growth factors including nerve-derived growth factor (NGF), BDNF, neurotrophins and glial cell-derived neurotrophic factor (GDNF). M2 microglia also release pro-survival factor progranulin and induce mannose receptor (CD206), which is found in inflammatory zone 1 (FIZZ1), arginase 1 (Arg1). In summary, M1 microglia induce inflammation and neurotoxicity, while M2 microglia induce anti-inflammatory and neuroprotection [31].

However, this binary M1/M2 concept has recently been debated. Transcriptome studies have shown that there is a continuum of different intermediate microglial phenotypes; thus, the M1/M2 paradigm is inadequate to provide an accurate description of microglia activation in vivo [31]. In fact, microglia are cell populations with functional heterogeneity, and an age- and region-dependent gene expression signature. Their morphology, ultrastructure, and molecular profile are dynamic and plastic, resulting in the coexistence of many different cell states closely associated with their diverse functions [23]. Key modifying factors that lead to microglial heterogeneous states include age, sex, circadian time, local CNS signals, and peripheral cues, such as changes in the microbiota [32].

### 2.3. Physiological Functions of Microglia

The varied morphologies and distinct physiological functions of microglia in both development and adulthood have been extensively reviewed [22,33]. As the brain’s first line of cellular defense against invading pathogens and other types of brain damage, microglia exhibit the typical function of macrophages, such as the phagocytosis of pathogens and cell debris and the production of inflammatory factors, contributing to CNS immune defense [34]. In addition, microglia play essential roles in regulating brain development and maintaining CNS homeostasis by participating in synapse remodeling, neurogenesis, neuronal function, angiogenesis, and myelination [33,34]. Surveillance, phagocytosis, and the capacity to release soluble factors are core properties through which microglia fulfil their important biological functions [23]. Microglial mobility (capability of moving around) and motility (capability of extending/retracting processes), determined by their microenvironment, are essential for their normal functions in both developing CNS and adult brain [28].

The physiological functions of microglia In developing CNS were systemically reviewed in a recent study [28]. In brief, microglia’s functions are (1) to clear cellular debris by phagocytosis; (2) to provide tropic support for cell survival, proliferation, and differentiation; (3) to support the formation, maturation, and elimination of synapses; and (4) to influence the development and re-modeling of the CNS vasculature. Here, we have mainly summarized the physiological role of microglia in the adult brain, focusing on surveillance, phagocytosis, synapse pruning, interaction with neurons, maintenance of the neurovascular unit (NVU) integrity, and blood-brain barrier (BBB) permeability, all of which directly or indirectly influence neuronal activity, synaptic plasticity, and memory processes (Figure 2). Microglia under normal physiological conditions not only modulate memory strength, but also memory quality and forgetfulness in adult mice [35].

#### 2.3.1. Surveillance

In healthy adult brains, microglia function to constantly monitor the entire CNS with highly branched processes that can extend up to 50 µm from the soma [36]. Microglia are able to complete detection of the entire brain parenchyma every 24 h [30]. Surveillance is one of the core properties helping microglia to achieve their important biological functions. For example, through surveillance, microglia detect which synapses are strong, weak or inactive, and determine which synapses to remove while others are maintained or strengthened. Surveilling microglia change their morphology in response to alterations in the surrounding microenvironment [37], monitoring neuronal health and function status through their necessary branches [9]. Neurons function to maintain microglial homeostasis by releasing cytokines and neurotransmitters [38]. In a healthy state, the maintenance of microglial homeostasis depends on a balance between endogenous and exogenous factors [39]. Endogenous factors mainly include runt-related transcription factor-1 (RUNX1) and interferon regulatory factor-8 (IRF-8). Exogenous factors mainly include CD200, C-X3-C motif chemokine receptor 1 (CX3CR1), triggering receptor expressed on myeloid cells 2 (Trem2), etc.

#### 2.3.2. Phagocytosis

Microglia protect the brain and maintain microenvironment homeostasis by removing harmful factors through phagocytosis. Microglia can actively remove dead neurons, proteins, and cell debris [40]. The specific process of phagocytosis is transporting the “eaten” harmful macromolecules to lysosomes for decomposition, similar to autophagy in neurons [41,42]. Microglia can also phagocytose surviving neurons, but only for those that are aged or damaged [43]. Microglia can phagocytose a large number of apoptotic neuroblasts in the subgranular region of the hippocampus in young mice, indicating that they are involved in neurogenesis [44]. During neurogenesis, microglia act as the primary phagocytic cells in the CNS, continuously removing neurons, synapses, and myelin [45] to regulate neuronal growth and development.

Microglia, as mobile phagocytes in the CNS, change their own state or phenotype dynamically, for example, from an activated state to a neurotoxic state [46]. The state changes of microglia are plastic and strongly depend on the context, such as the type and extent of CNS damage [47]. Once there are deleterious factors that disrupt CNS homeostasis, the dynamic mechanisms to activate microglia are triggered [48]. This activation process involves the proliferation and migration of microglia to the injury site, morphological changes such as transition from a highly branched state to an amoeboid state, and profile changes in secretion [49].

Microglia activation plays a dual role. On one hand, activated microglia produce a variety of neurotoxic pro-inflammatory mediators resulting in inflammation, such as IL-6, NO, and TNF-α [50]. On the other hand, neuroimmune regulatory microglia secrete a variety of neurotrophic factors, such as IGF-1, BDNF, and TGF-β [51], which help to end the inflammatory process and promote tissue repair [52]. After inflammation is eliminated, microglia resume their surveillance function [9]. Microglia are usually considered to polarize toward the M1 phenotype under initial insults, producing and releasing pro-inflammatory factors, followed by polarization towards the M2 phenotype, involving active phagocytosis and the release of anti-inflammatory factors [53]. However, recent studies using single-cell technologies provide clear evidence that microglia in the living brain often co-express M1 and M2 molecular signatures. In addition, M2 microglia do not always play a neuroprotective role [54]. Therefore, the M1-M2 dichotomy is too simplistic to cover all microglial phenotypes or states [55].

#### 2.3.3. Synapse Pruning and Elimination by Phagocytosis

Microglia can identify and directly remove unnecessary synapses from neurons during development [56,57] and throughout adulthood [9]. In the healthy brain, microglia prune away weak or unwanted synaptic structures by making brief, repeated contacts with synapses, promoting the refinement of neural circuits and increasing neuronal network efficiency [58]. Studies show that the elimination of microglia in adult mice elevates the expression of synaptophysin and post-synaptic density 95 (PSD95) and increases dendritic spine density [59].

The precise mechanism of microglia-mediated synapse pruning is largely unknown; however, recent studies have identified specific signaling pathways regulating the recruitment of microglia to certain synapses for pruning. First, the classical complement cascades function as a “tagging” signal for synapse pruning. Complement proteins C1q and C3 are predominantly produced by microglia or astrocytes and bind to apoptotic, immature, or weak synapses [60]. Microglia recognize C1q- and C3-tagged synapses via the exclusive complement receptor CR3, triggering synapse engulfment in development and adulthood [61,62]. Phosphatidylserine (PS) is another “Eat me” signal promoting synaptic pruning. TREM2, a cell surface receptor on microglia, can recognize PS typically exposed on apoptotic or injured dendrites, and consequently, induce synapse elimination [63]. The interaction between PS and TREM2 regulates synaptic pruning in the hippocampus and dorsal lateral geniculate nucleus in vivo and in vitro [64]. Microglia also regulate synapse elimination through highly expressed CX3CR1 binding to neuronal CX3CL1. The CX3CR1/CX3CL1 signaling is essential for microglia–neuron communication, allowing microglia to know which synapses need to be eliminated [65]. In addition, it also regulates the microglial engulfment of oligodendrocyte progenitor cells (OPCs) during development [35].

In contrast to “Eat me” signals, “Don’t eat me” signals act to inhibit synaptic pruning, and thus, protect strong and active synapses from being engulfed. In particular, CD47 is a “Don’t eat me” protein localized on highly active synapses, and its receptor SIRPα is highly expressed on microglia during peak pruning periods in development to suppress the phagocytosis of synapses. In apoptotic or damaged cells, the “Don’t eat me” signals are downregulated, which enables the “Eat me” signal to engulf the damaged cells [66,67]. The neuronal CD200 interacting with microglial CD200R may be another “Don’t eat me” signal [68]. Overall, synaptic pruning by microglia is a highly regulated process, which begins in development and occurs throughout adulthood in multiple brain regions, playing a key role in regulating normal neuronal activity, synaptic plasticity, learning, and memory [35].

#### 2.3.4. Interaction between Neurons and Microglia

In physiological conditions, motile microglial processes are constantly monitoring their microenvironment. Through neuron and microglia interactions, microglia are able to regulate neuronal activity, synapse formation and survival, and synaptic circuit remodeling [35,48]. In particular, microglia play a significant role in setting the synaptic tags during the early phase of activity and plasticity, thus, promoting the stability of long-term potentiation (LTP) under normal physiological conditions [69]. The interaction between neurons and microglia is currently known in three ways:

First, direct physical contact. The motile processes of microglia physically contact synapses and modulate neuronal activity [70]. The frequent contact between microglia and synapses in the healthy brain promote local network synchronization, followed by an increase in neuronal activity [71]. Studies have also shown that a subpopulation of microglia can specifically bind to the axon initial segment of cortical neurons in the healthy brain, which disappear following brain injury and inflammation [72]. Neuronal soma has recently been identified as a site of microglial interaction, relying on purinergic signaling and possibly involved in microglia-induced neuroprotection [73]. Nodes of Ranvier is a new site for microglia–neuron communication in mouse and human CNS, which depends on neuronal activity and potassium ion release and contributes to remyelination [74].

Second, via ligand/receptor binding and signaling. Ligands produced by neurons can bind to the receptors specifically expressed on the surface of microglia [75]. For example, CD200 secreted by neurons specifically binds to CD200R on the surface of microglia, suppressing microglial activation and facilitating synaptic plasticity [76]. Microglia-neuron interactions are able to prevent excessive neuronal activation via the adenosine triphosphate (ATP)-AMP-adenosine (ADO)-A1R-dependent negative feedback mechanism [48]. Microglial processes sense locally released ATP following synaptic activation through purinergic receptor P2RY12, which controls ATP/ADP-dependent chemotaxis and motility. Meanwhile, microglia break down ATP using enzymes CD39 and CD73 to produce adenosine, which acts on the adenosine receptor A1R localized on active neurons to prevent neuronal hyperactivation. Studies also show that the interaction between neuronal CX3CL1 and microglial CX3CR1 promotes cytokine release from microglia and enhances synaptic plasticity [77,78]. When the CNS is pathologically injured, CX3CL1 expression is upregulated, and the number of microglia surrounding injured neurons increases [79].

Third, via soluble factors and cytokines released by microglia. Microglia exhibit variable effects on synaptic plasticity depending on the dynamic release of cytokines, such as PI3K, BDNF, CREB, TNFα, and IL-1β. For example, microglia can enhance synaptic plasticity by the expression and secretion of BDNF via microglial phosphatidylinositol 3-kinase (PI3K)/CREB/BDNF signaling. Mature BDNF then binds to TrkB receptors on neurons and triggers downstream signaling cascades [80]. In contrast, microglia released IL-1β and TNFα inhibit synaptic plasticity [81].

Fourth, via soluble factors released by neurons. For example, the experience-dependent expression of IL-33 in adult hippocampal neurons induces microglial engulfment of the extracellular matrix, a structure involved in neuronal plasticity and memory. Removal of the IL-33 receptor specifically in microglia reduces newborn neuron integration, causing impaired memory precision [82]. IL-34 secreted by neurons binds to CSF-1R on microglia to enhance their proliferation, survival, and function [83]. ATP local release by activated neurons mediates microglial process motility via binding to microglial P2Y12 receptor. Neurons also affect the function of microglia by releasing neurotransmitters. For example, norepinephrine (NE)/microglial β2-adrenergic receptor (β2-AR) signaling is a critical regulator for microglia surveillance and injury response in the visual cortex of awake mice [84].

#### 2.3.5. Maintenance of the NVU Integrity and the BBB Permeability

The NVU is a complex functional and anatomical structure composed of neurons, glial cells, vascular cells, and a basal lamina formed by brain endothelial cells and extracellular matrix. The NVU components are closely linked to each other, enabling an efficient system to support cerebral blood flow, neuronal metabolic activity, and BBB permeability that serves to maintain the homeostatic microenvironment of the CNS and protect the brain from exposure to pathogens and toxic agents. Dysregulation of the NVU and BBB are critical pathophysiological events in neurodegenerative diseases, including AD [85].

Microglia are an important component of the NVU, which actively communicate with the endothelium and regulate the BBB integrity and permeability. When an injury occurs, such as stroke, microglia are activated and develop into a spectrum of different phenotypes with overlapping functions, including pro-inflammatory M1 and anti-inflammatory M2 microglia. Activated microglia then play dual roles at the BBB, depending on their functional states. On the one hand, they produce high levels of pro-inflammatory cytokines (such as IL-1β, TNF-α, IL-6, NO, iNOS), causing increased permeability and damage of the BBB. One the other hand, activated microglia can phagocytose cellular debris and release anti-inflammatory cytokines (such as IL-4, IL-10, TGF-β) and growth trophic factors (such as vascular endothelial growth factors VEGF, BDNF), which promotes angiogenesis and facilitates BBB repair [86].

## 3. Microglial Mechanisms of AD Pathogenesis and Memory Impairment

Microglia are keen responders and critical players in numerous disease conditions, responding to various challenges by changing their molecular profile, morphology, ultrastructure, as well as motility and function [23]. While microglia play a beneficial role in maintaining CNS homeostasis and regulating neuronal activity, synaptic plasticity, and cognitive function, they can also be detrimental in disease conditions or stages. Microglia-mediated neuro-inflammation is a common feature of neurodegenerative diseases including AD, Parkinson’s disease (PD), amyotrophic lateral sclerosis (ALS), and multiple sclerosis (MS). Studies demonstrate that microglia dysfunction contributes to the progression of AD and cognitive decline, and IL-33 can increase microglia phagocytic function while decreasing inflammation in the brain, ultimately improving synaptic plasticity and memory [87]. Here, we highlight the present understanding of microglial mechanisms underlying AD-associated memory impairment through reviewing and summarizing recent studies indicating a causal relationship between microglia dysfunction and AD progression (Figure 3).

### 3.1. Neuro-Inflammation Mechanism

A large number of studies have proven that neuro-inflammation and microglial activation appear several years before the onset of AD [88,89]; therefore, it is believed that neuro-inflammation is a key factor in AD pathogenesis and the third pathological hallmark of AD, in addition to Aβ and NFT [40]. The expression of inflammatory factors and chemokines was upregulated in AD patient brains [90]. Neuro-inflammation is also closely associated with the accumulation of Aβ and tau proteins [89]. The fact that the hippocampus is the most vulnerable brain region to neuro-inflammation and AD pathology provides spatial evidence of the association between neuro-inflammation and AD memory impairment [91]. Neuro-inflammation causes neuron dysfunction and cell death both directly and indirectly. Microglia and other immune cells in the CNS directly release neurotoxins to damage and phagocytose neurons. Meanwhile, disrupted homeostasis, such as Aβ and tau deposits, prompts microglia and astrocytes to release inflammatory factors and chemokines, resulting in neuronal death [92].

Microglia are the major mediators of neuro-inflammation. They develop towards the pro-inflammatory M1 phenotype after sensing damage-associated molecular patterns (DAMPs), such as Aβ and tau protein in AD, through pattern-recognition receptors (PRRs) highly expressed in microglia. Signaling through microglial PRRs—such as Toll-like receptors (TLRs)—initiate a rapid response, including cell migration, proliferation, and the release of pro-inflammatory factors, chemokines, neurotrophic factors, and neurotoxins that damage neurons [93,94]. In the early stage of AD, Aβ activates microglia to secrete pro-inflammatory factors, leading to chronic persistent inflammation in the brain parenchyma [40]. After activation, microglia further release pro-inflammatory factors, such as TNF-α and IL-1β. TNF-α aggravates the inflammatory response in AD [95] and disrupts the homeostasis of synaptic connections [96]. IL-1β can mediate the toxic effects of Aβ on synapses, causing changes in synaptic plasticity [97]. While microglial activation is beneficial In preventing AD-associated pathology, the chronic activation of microglia is detrimental, as prolonged TLR2 and TLR4 activation in microglia induces Aβ production [98]. When activated, neuro-inflammatory microglia also provoke the formation of A1 reactive astrocytes by simultaneously secreting IL-1α, TNF, and C1q. A1 astrocytes express C3; lose the ability to promote neuronal survival, outgrowth, synaptogenesis and phagocytosis; but gain a new neurotoxic function by the secretion of a soluble neurotoxin, rapidly killing neurons and mature, differentiated oligodendrocytes. A1 astrocytes are abundant in AD and contribute to the death of neurons and oligodendrocytes [99].

It is believed that oxidative stress is a consequence of neuro-inflammation in AD pathogenesis [100]. Oxidative stress can directly induce neuronal death, and the pro-inflammatory activation of microglia further exacerbates oxidative stress [101]. Microglia-mediated clearance mechanisms do not appear to participate in this process [102]. Moreover, neuro-inflammation impairs neurotrophic factor-induced neuroprotection [103]. For example, neuro-inflammation promotes the pathological development of AD by interfering with the maturation, and thus, the function of NGF [104]. NGF is a neurotrophic factor necessary for neuronal survival and homeostasis, and its dysfunction leads to cognitive decline in AD patients [105,106]. Neuro-inflammation can also over-activate matrix metalloproteinase-9 (MMP-9), leading to NGF dysfunction [107].

When activated, innate immune cells in the CNS, mainly microglia and astrocytes, can cause programmed cell death (PCD) through multiple pathways, including pyroptosis, apoptosis, necroptosis, and PANoptosis, which is an inflammatory cell death integrating components from other three pathways. Pyroptosis and necroptosis are highly pro-inflammatory due to the release of cell contents, which can induce severe inflammation. The pyroptosis pathway involves inflammasome formation, activation of caspase-1, and the release of pro-inflammatory cytokines IL-1β and IL-18. Necroptosis occurs in response to caspase-8 inhibition and is receptor-interacting serine/threonine-protein kinase (RIPK)- and mixed lineage kinase domain-like pseudokinase (MLKL)-dependent. Although apoptosis is a typically non-inflammatory cell death pathway initiated by activation of caspase-8, followed by caspase−9 and caspase-3, it can cause the release of inflammatory mediators when upregulated or when cell debris is improperly cleared. Pro-apoptotic B-cell lymphoma-2 (Bcl-2) family members are markers of apoptotic cell death. In chronic disease conditions such as AD and other neurodegenerative disease, cell death can be both a consequence and a cause of inflammation [98,108,109].

Overall, neuro-inflammation associated with AD is a chronic, self-sustaining response that may trigger abnormal brain function [110]. The release of pro-inflammatory factors and chemokines may promote the perpetuation of inflammatory responses in the CNS, switching microglial activation into a hyperactive state. A further release of inflammatory factors and neurotoxins leads to synaptic loss and neuronal death [111] and prompts the development of AD pathology and memory decline.

### 3.2. Microglial Dysfunction Mechanism

Evidence supporting a key role of microglial dysfunction in AD pathogenesis first comes from genetic studies. Trem2 is considered an important innate immune receptor in microglia that is closely related to Aβ pathology [112] and mediates the phagocytic clearance of neuronal debris. Trem2 mutation causes the impaired clearance of Aβ by microglia [113]. Trem2 also negatively regulates inflammation [114]. Single-cell RNA sequencing identified a new subset of microglia associated with AD and aging—disease-associated microglia (DAM) [115]. Trem2 is required for DAM formation and activation [116]. In addition, it has been reported in the literature that a rare mutation in Trem2 (R47H) can prevent microglia from clearing Aβ, resulting in the continuous accumulation of Aβ and aggravation of AD progression [117].

Second, there is evidence that changes in microglial telomeres are closely related to dementia [118]. The microglia telomeres of aging rats are significantly shortened and have decreased activity. Dystrophy of microglial telomeres is dramatic in AD, indicating reduced activity [118].

Third, deficiency in Aβ phagocytosis and clearance. Studies have demonstrated that activated microglia in AD sense and phagocytose Aβ through cell surface receptors such as CD36, CD47, CD14, and TLRs. In microglia, endocytosed Aβ is loaded into phagocytic vesicles that merge with lysosomes to form phagosomes [119] for degradation and elimination [120]. Changes in the formation or function of phagosomes may cause increased accumulation of intracellular debris, leading to cell death [121]. However, as AD progresses, receptors mediating Aβ phagocytosis are continuously downregulated, along with a reduced activity of Aβ-degrading enzymes, resulting in a deficiency in Aβ clearance by microglia [122]. Accumulated Aβ further drives neuro-inflammation and accelerates the pathological development of AD by producing a large number of pro-inflammatory mediators and neurotoxins [123]. Meanwhile, accumulated Aβ in cells leads to microglial death and the release of non-degraded Aβ [124], further deteriorating neuro-inflammation and the formation of Aβ plaques in the brain. Studies find that Aβ inhibits the phagocytic ability of microglia in AD [125]. Aging-induced microglial dysfunction may be a key factor affecting the efficiency of Aβ plaque phagocytosis and clearance [126]. Aging inhibits the clearance of Aβ by microglia, and young microglia can increase Aβ clearance by senescent microglia [41].

In conclusion, microglial dysfunction associated with AD may result from genetic mutation or Aβ stimulation [127]. Dysfunctional microglia will lose their beneficial roles, such as phagocytosis of DAMPs or pathogen-associated molecular patterns (PAMPs); meanwhile they exhibit harmful phenotypes, such as releasing a large number of neurotoxins [128], and gradually become ineffective in Aβ clearance. Indeed, studies show that IL-33 administration can increase microglia phagocytic function and decrease inflammation in the brain, which ultimately improves synaptic plasticity and memory in APP/PS1 mice [87].

### 3.3. Pathological Synaptic Loss and Dysfunction Mechanism

The hypothesis of pathological synapse loss suggests that the main cause of AD memory impairment and cognitive decline may not only be neuronal loss and the accumulation of Aβ plaques and NFTs, but synaptic loss and synaptic dysfunction also make a significant contribution [129]. Synaptic dysfunction precedes Aβ plaque and NFTs pathology as well as cognitive impairments in AD [130]. Studies on AD patients have also shown that synaptic changes are closely related to the severity of cognitive impairment [131]. Cognitive decline observed in AD model mice without neuronal death supports this statement [131]. Microglia and Aβ are likely involved in synaptic loss and dysfunction.

#### 3.3.1. Synapse Loss Associated with Microglia

Microglia eliminate unnecessary synapses through C1q- and C3-mediated phagocytosis during development and throughout adulthood [58]. Aβ plaques and NFTs can induce microglial activation and complement cascades-dependent synapse clearance, which in turn promotes AD pathology [132]. In the hippocampus of AD model mice, complement C1q and C3 are upregulated and linked to synapses, resulting in the increased phagocytosis of synaptic elements by microglia and a reduced number of synapses [133]. Inhibition of C1q, C3, and CR3 rescues synaptic loss and cognitive dysfunction in AD model mice, further supporting that the complement-mediated microglial phagocytosis of synapses is involved in AD synaptic loss [129].

In addition, microglia can shape synapses by direct contact with them [134]. Following synaptic injury, microglia tightly associate with dendrites and directly separate pre-synaptic and post-synaptic neurons, a process known as “synaptic stripping” [135]. Microglia-involved synaptic stripping depends on the activity of neurons [136], and the stripping time is prolonged due to ischemia, leading to an increased probability of synapse loss following contact [137]. The specific mechanisms of synaptic stripping and associated microglial subtypes are currently unknown [134].

#### 3.3.2. Aβ-Induced Synaptic Toxicity and Dysfunction

A number of studies support that AD is the result of synaptic failure, and that soluble Aβ oligomers (AβOs) cause AD synaptic and behavioral pathogenesis [138]. AβOs levels are markedly elevated in the early stages of AD, localizing at or within the synapse, and inversely correlate with synaptic loss [139]. The precise mechanism of AβOs-induced synaptic toxicity and dysfunction remains uncertain. Studies postulate over 20 possible receptors involved in the toxicity of AβOs, including glutamate, adrenergic, acetylcholine receptors, and others. Unfortunately, no single candidate receptor protein has yet proved responsible for all features of AβO activity [140]. In addition, AβOs are capable of targeting multiple synaptic proteins outside and inside the synapse, including α3-Na/K-ATPase on synaptic membranes, and synGap and Shank3 in post-synaptic density (PSD), contributing to synaptic dysfunction and loss in early stages of AD [141]. AβOs also trigger synaptic loss at the level of the dendritic spines via upregulation of RAPGEF2 and activation of the Rap2-JNK pathway in the hippocampus of AD model mice [142].

### 3.4. Microglia Drive Tau Pathology and Dissemination

In the healthy brain, tau protein is an abundant microtubule-binding protein involved in the formation and stabilization of microtubules and plays an important role in axonal transport [143]. Tau protein is mainly expressed in mature neuron axons, and its phosphorylation is necessary for normal physiological functions [144]. In AD, the binding of tau to microtubules is disrupted, resulting in a massive increase in free tau and aggregation into NFTs [145]. NFT density is positively correlated with the severity of cognitive impairment in AD patients, indicating that insoluble tau protein is neurotoxic [146]. NFTs cause abnormal accumulation of proteins by downregulating proteasome activity, leading to neuronal death [147]. Post-synaptic redistribution of pathological tau protein found in AD patient brains may also be associated with neurotoxicity [148]. In addition, in AD, not only is the secretion process of tau protein more active [149], but tau protein can also undergo various post-translational modifications, such as phosphorylation, methylation, and glycosylation. Hyperphosphorylation of tau protein has been widely studied and is considered closely related to the development of AD pathology.

Tau appears to be more strongly associated with cognitive decline than Aβ [150]. Hyperphosphorylated tau itself can lead to reduced dendritic spine numbers and synaptic damage [151]. Hyperphosphorylated tau localizes both pre- and post-synapse, leading to synaptic dysfunction by impairing synaptic transmission and excitability [152,153]. Tau knockout mice are immune to NMDA receptor excitotoxicity and Aβ-induced neuronal damage [154], indicating that tau protein is essential for neuronal damage. Increasing Trem2 was found to inhibit tau hyperphosphorylation in Aβ-induced AD model mice [155]. Trem2 signaling has also been associated with tau pathology and synaptic loss in AD patient brains [148].

Animal studies have demonstrated that microglial activation precedes NFT formation [156]. Microglia interact with and internalize extracellular tau via CX3CR1 [144]. The knockdown of CX3CR1 in tau-induced AD model mice altered microglial activation and behavioral abnormalities caused by tau hyperphosphorylation [157], providing a link between microglial activation and tau pathology. Microglia activation not only induces tau hyperphosphorylation by secreting a large number of inflammatory factors, but also promotes the proliferation of hyperphosphorylated tau in the CNS, indicating that microglia drive tau pathology [94].

The relationship between microglia and tau goes far beyond this. When CSF1R inhibitors are used to eliminate microglia, the production of tau is inhibited, indicating the importance of microglia in tau formation [158]. In AD, IL-1 secreted by activated microglia promotes tau hyperphosphorylation and aggregation in neurons through a TLR-4-mediated inflammatory pathway [157].

### 3.5. Pathological NVU Dysfunction in AD

In the AD brain, Aβ deposits both in the brain parenchyma as neuritic plaques and around blood vessels as cerebral amyloid angiopathy (CAA). CAA is characterized by Aβ deposition within the walls of cortical and leptomeningeal vessels, and perivascular microglial activation, which leads to NVU dysfunction. Studies show that perivascular microglia activation promotes blood vessel disintegration in the ischemic penumbra [159]. CAA is universally found in AD brains and has been linked to neuro-inflammation, chronic hypoperfusion, ischemia, loss of the blood vessel wall integrity, and hemorrhage. Pathological NVU dysfunction has emerged as an essential element in AD pathogenesis [160].

## 4. Clinical Implication

Clearly, innate immune activation and neuro-inflammation play a prominent role in the pathogenesis and progression of AD. Since microglia are the primary players in neuro-inflammation, modulating their activation could be a therapeutic strategy for AD and associated cognitive decline. At present, there are three microglia-based treatments for AD memory impairment, considering the important role of microglial activation in AD pathogenesis.

### 4.1. Inhibition of Neuro-Inflammation

AD neuro-inflammation is mainly attributed to microglia [161], so drugs used to treat neuro-inflammation are designed to inhibit their pro-inflammatory activation. The most common drugs are non-steroidal drugs (NSAIDs) [104]. Suppressing neuro-inflammation with NSAIDs was one of the earliest approaches to treat AD [161], and NSAIDs were found to reduce the number of microglia in AD patient brains at autopsy [162]. However, NSAIDs have no significant effect on cognitive decline in patients with AD [104], suggesting that non-specific inhibition of neuro-inflammation may not be an effective strategy for AD treatment. Interestingly, minocycline, which has anti-inflammatory effects by inhibiting microglial activation, was found to reverse Aβ-induced memory impairment in animal studies [163], whereas it did not improve cognitive deficits in patients with mild AD [164].

Attempts have been made to target the NF-κB, NLPR3–caspase 1, and p38MAPK pathways, thus, suppressing the pro-inflammatory microglial phenotype, or even more specifically, to target downstream pro-inflammatory cytokines, such as TNFα and IL-1. For example, the caspase 1 inhibitor VX-765 shows some potential to suppress microglial activation, Aβ deposition, and neurodegeneration, and to alleviate cognitive decline in a mouse model of AD. Those findings highlight that therapeutic strategies that specifically target pro-inflammatory microglia could be more effective than counteracting neuro-inflammation in general. However, compounds for clinical application are still under development or evaluation for safety and efficacy [161].

### 4.2. Regulating Microglial Phenotype and Improving Microglial Dysfunction

Microglia activation in the very early stages of AD is known to be protective by clearing toxic damage and maintaining neuronal function. Therefore, transforming microglial activation toward an anti-inflammatory/phagocytic phenotype was promoted to alleviate AD pathology in preclinical studies. For example, various anti-inflammatory cytokines, including IL-33 and IL-4, have shown the potential to modulate microglial activation and alleviate AD pathology. In addition, peroxisome proliferator-activated receptor-γ agonists (such as pioglitazone and rosiglitazone) were confirmed to be able to facilitate the transformation of microglia from a pro-inflammatory to a phagocytic phenotype and enhance the clearance of Aβ [165]. However, rosiglitazone showed no significant effect in a phase III trial, probably owing to its poor BBB permeability and safety concerns [166]. Moreover, studies report that inhibiting the mTOR pathway could significantly reduce Aβ-induced microglial toxicity [127]. However, Aβ plaques were found to be increased in 5 × FAD mouse brains after treatment with metformin, a commonly used mTOR inhibitor [167]. Altogether, strategies aimed at modulating microglial activation seem encouraging; however, the efficacy in humans largely remains to be evaluated.

### 4.3. Intervention of Microglia Priming

Microglial priming is associated with both aging and neuro-inflammation, and primed microglia show enhanced sensitivity and reactivity to inflammatory stimuli. Thus, interference with microglial priming at the early stages of AD may be an effective therapeutic strategy. Midlife obesity, insulin resistance, and elevated low-density lipoprotein (LDL) levels have been suggested to induce inflammation and microglial priming. Recent studies have reported that interfering with microglial priming with a multifaceted approach does have certain positive effects on cognition, which include statins, folic acid, omega 3 fatty acids, cognitive training, physical exercise, nutrition intervention, and others [161,168,169,170]. However, whether these multidomain interventions can prevent or diminish AD pathogenesis and memory impairment is still uncertain.

In summary, interventions to target microglial priming in the pre-disease period and to modulate microglial states in the early stages of AD have shown promise as part of a disease-modifying strategy. However, caution should be taken when targeting microglia, since their activation is context-dependent, exhibiting diverse phenotypes and probably having multifaceted influences on the progression of AD. For example, TREM2 is an attractive target for the pharmacological modulation of microglial activation. TREM2 signaling has been suggested to promote microglial proliferation, phagocytosis and cytokine secretion, and to regulate microglial metabolism and survival. Nevertheless, since TREM2 is a vital upstream mediator of microglial activation, changes in TREM2 function might have an effect on both beneficial and detrimental microglial functions in AD [161].

## 5. Conclusions

AD is one of the major medical and social problems without effective treatment worldwide. Many recent studies have disclosed the critical role of microglial activation in the development of AD pathology. Microglia activation in AD is dynamic and complicated, ranging from progressive activation to partial activation to full activation. In the early stage of AD, microglial activation can phagocytose Aβ to inhibit Aβ deposition, and thus, achieve neuroprotective effects [18]. With AD progression, chronically activated microglia gradually lose the capability or efficiency to clear Aβ [169], while they continuously produce pro-inflammatory and neurotoxic factors, leading to neurodegeneration. Most recently, microglia-based therapeutic strategies did not show significant improvement on AD memory impairment, although each specific aspect of microglia activation could be selectively regulated, such as migration, phagocytosis, or cytokine release. The contribution of microglial activation in AD does not appear to be straightforward. Further studies are still required to illustrate the complicated mechanism underlying the involvement of microglia in AD pathogenesis. Targeting microglia may provide a potential strategy for the accurate treatment of memory impairment associated with AD.

## Figures and Tables

**Figure 1 biomolecules-12-01722-f001:**
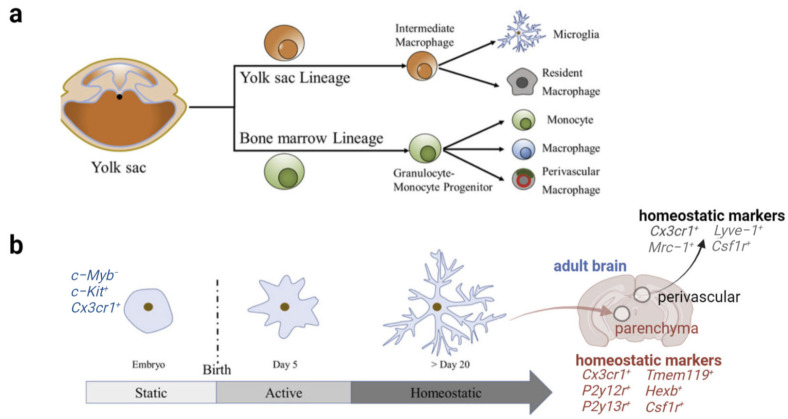
The origin of microglia and morphological changes during development and throughout adult mouse brain. (**a**) Microglia originate from the yolk sac and eventually mature following the development of intermediate macrophages. (**b**) Immature microglia from the embryonic period develop into homeostatic microglia 28 days after birth. Immature microglia exhibit amoeboid-like morphology. Matured microglia show homeostatic, highly ramified morphology. Microglia in different states show distinct molecular signatures. Abbreviations: *c–Myb*, protein c myeloblastosis gene; *c-Kit*, receptor tyrosine kinase gene; *Cx3cr1*, C-X3-C motif chemokine receptor 1 gene; *P2y12r*, purinergic receptor *P2Y12* gene; *P2y13r*, purinergic receptor P2Y13 gene; *Tmem119*, transmembrane protein 119 gene; *Hexb*, beta-hexosaminidase subunit beta gene; *Csf1r*, colony-stimulating factor 1 receptor gene; *Mrc–1*, mannose receptor c-type 1 gene; *Lyve-1*, lymphatic vessel endothelial hyaluronan receptor 1 gene.

**Figure 2 biomolecules-12-01722-f002:**
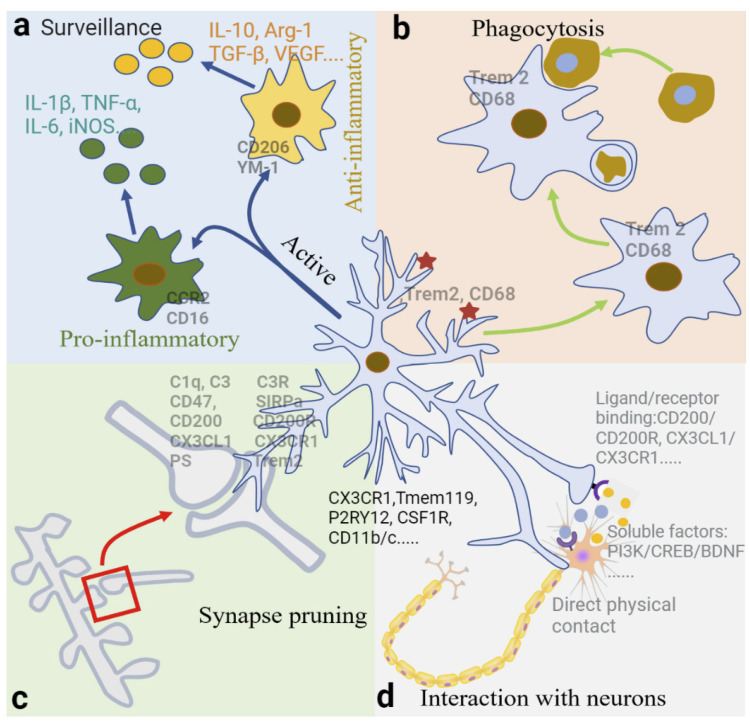
Physiological function of microglia. (**a**) Microglia constantly monitor neuronal health and function status through their necessary branches. Microglia change their morphology in response to alterations in the surrounding microenvironment. Activated microglia can exert pro-inflammatory or anti-inflammatory roles by secreting cytokines. The surface markers and the products secreted by each phenotype are shown in the figure. (**b**) Microglia protect the brain and maintain microenvironment homeostasis by removing harmful factors through phagocytosis. (**c**) Microglia execute synaptic pruning. Signaling involved in microglia-mediated synaptic pruning is listed in the figure. (**d**) Microglia interact with neurons directly or indirectly, via direct physical contact, ligand/receptor binding and signaling, and soluble factors released by microglia and neurons. Abbreviations: IL-1β, interleukin-1β; IL-6, interleukin-6; IL-10, interleukin-10; TNF-α, tumor necrosis factor α; iNOS, inducible nitric oxide synthase; Arg-1, arginase 1; TGF-β, transforming growth factor β; VEGF, vascular endothelial growth factor; CCR2, C-C motif chemokine receptor 2; CD16, cluster of differentiation 16, the Fc receptors FcγRIIIa; CD206, cluster of differentiation 206, the mannose receptor; YM-1; chitinase-like protein 3, Chil3; Trem2, triggering receptor expressed on myeloid cells 2; CD68, cluster of differentiation 68; C1q and C3, the complement component 1q and 3; C3R, C3 receptor; CD47, cluster of differentiation 47; SIRPa, signal regulatory protein α; CD200, cluster of differentiation 200; CD200R, CD200 receptor; CX3CL1, C-X3-C motif chemokine ligand 1; CX3CR1, C-X3-C motif chemokine receptor 1; PS, phosphatidylserine; Tmem119, transmembrane Protein 119; P2RY12, purinergic receptor P2Y12; CSF1R, colony-stimulating factor 1 receptor; CD11b/c, cluster of differentiation 11b/c; PI3K, phosphoinositide 3-kinases; CREB, cAMP response element binding protein; BDNF, brain-derived neurotrophic factor.

**Figure 3 biomolecules-12-01722-f003:**
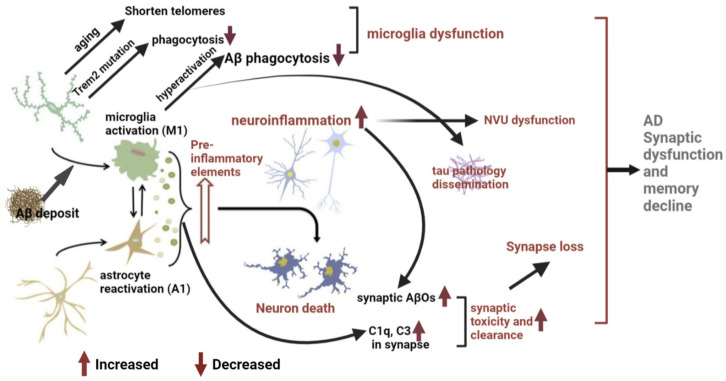
Microglial mechanism of memory impairment in AD. Chronic and pathological activation of microglia during AD progress causes neuro-inflammation, microglia dysfunction, synapse loss and synaptic toxicity, and microglia-driven tau pathology, which work together to contribute to AD synaptic dysfunction and memory impairment. Abbreviations: Aβ, Amyloid β; Trem2, triggering receptor expressed on myeloid cells 2; AβOs, soluble Aβ oligomers; C1q and C3, the complement component 1q and 3; NVU, neurovascular unit.

## Data Availability

Not applicable.

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
