# Peer review of "New Insights into Microglial Mechanisms of Memory Impairment in Alzheimer’s Disease"

_biomolecules, 2022, doi:10.3390/biom12111722_

Round 1

Reviewer 1 Report

The review article by Na Li et al. integrates the physiological and pathophysiological roles that contribute to memory dysfunction in Alzheimer's disease. The article is well-developed and integrates the important roles of microglial cells in this neurodegenerative disorder. However, I suggest that the authors make the following recommendations to improve the manuscript:

Major concerns

1. The authors need to discuss the subtopics developed throughout the manuscript.

2. In the section "2.1 Source of microglia" it is necessary to detail the importance of immature microglia's morphological and functional changes from the embryonic period into homeostatic microglia 20 days after birth.

3. In the section “2.2.1. Supervision” add more information and discuss the physiological relevance of microglia mobility and motility (10.1016/j.pneurobio.2019.04.001).

4. It is important that in another sub-section different from 2.2.2 "phagocytosis", they develop the M1 and M2 polarization state of microglia, briefly including the signaling pathways involved, surface markers that differentiate one phenotype from the other and subclasses of phenotypes identified.

5. In addition to the physiological and pathophysiological roles of microglia developed throughout the manuscript, it is essential that they briefly describe and discuss the importance of perivascular microglia and their impact on the maintenance of the BBB and the neurovascular unit. Add the following manuscripts: 10.1007/s00401-014-1372-1; https://pubmed.ncbi.nlm.nih.gov/33670754/

6. It would be interesting if the authors discussed cell death by inflammation or pyroptosis, in which microglia may play an important role (paragraphs 220-226).

7. The article could increase its impact if Figures 2 and 3 are improved:

• In Figure 2A, schematize the morphological differentiation of microglia, their surface markers, and the products secreted by each phenotype.

• Figure 3 does not include insights or novel mechanisms through which microglia lead to synaptic dysfunction and memory impairment. The authors should improve this figure by adding the mechanisms described in the text: 1) microglial dysfunction in AD pathogenesis (genetic studies; TREM 2); 2) changes in microglial telomeres; 3) phagocytose Aβ through cell surface receptors; 4) aging-induced microglial dysfunction and its relationship with early synaptic loss (via C1q- and C3-mediated phagocytosis).

• Incorporate in figure 3 and discuss in the text (could be included between lines 260 or 274) the possible activation of microglia by external stimuli that can mediate the activation of A1 or neurotoxic astrocytes and both lead to neuronal death                             (https://www.nature.com/articles/nature21029).

• Make another figure or scheme adding the possible therapeutic targets on microglia activation (described in the manuscript) and their effect on synaptic dysfunction, for example: Inhibition of C1q, C3, and CR3; nonsteroidal drugs; minocycline, peroxisome proliferator-activated receptor-γ inhibitors; statins, folic acid, omega 3 fatty acids, cognitive training, physical exercise, nutrition intervention, etc. On the other hand, in the text discuss the adverse effects that could be triggered by inhibiting microglia and indirectly certain physiological roles.

8. The authors mention that the role of the interaction between neurons and microglia (L185-186) is not clear. I suggest you review the following reference and discuss it: https://www.nature.com/articles/s41467-021-25486-7)

Minor concerns

1. In the abstract, the phrase “new therapeutic” is repeated in the same sentence, change it to another.

2. In keywords change AD for Alzheimer’s Disease

3. In the introduction, highlight that AD is the most common neurodegenerative disorder worldwide

4. Define Trem2, CD33, and ABCA7 (line 65)

5. In L82 in the same sentence “different” is repeated

Author Response

Thanks for all reviewers’ comments. Here we provide, point by point, the details of the revisions to the manuscript and our responses to the referees’ concerns.

Reviewer #1:

Major concerns

  1. The authors need to discuss the subtopics developed throughout the manuscript.

Our response: We added discussion about the subtopics according to the reviewer’s suggestion. See lines 72-75 of the revised manuscript.

  1. In the section "2.1 Source of microglia" it is necessary to detail the importance of immature microglia's morphological and functional changes from the embryonic period into homeostatic microglia 20 days after birth.

Our response: Thanks for the reviewer’s comments. Accordingly, we described the importance of morphological and functional changes of microglia from the embryonic period into adult 20 days after birth in our revised manuscript.

  1. In the section “2.2.1. Supervision” add more information and discuss the physiological relevance of microglia mobility and motility (10.1016/j.pneurobio.2019.04.001).

Our response: Thanks for the reviewer’s critical and instructive comments. Accordingly, we added more information and discussion about the physiological relevance of microglia mobility and motility in our revised manuscript.

  1. It is important that in another sub-section different from 2.2.2 "phagocytosis", they develop the M1 and M2 polarization state of microglia, briefly including the signaling pathways involved, surface markers that differentiate one phenotype from the other and subclasses of phenotypes identified.

Our response: Thanks for the reviewer’s critical and instructive comments. We added this section in revised manuscript according to the reviewer’s suggestion.

  1. In addition to the physiological and pathophysiological roles of microglia developed throughout the manuscript, it is essential that they briefly describe and discuss the importance of perivascular microglia and their impact on the maintenance of the BBB and the neurovascular unit. Add the following manuscripts: 10.1007/s00401-014-1372-1; https://pubmed.ncbi.nlm.nih.gov/33670754/

Our response: We do appreciate the reviewer’s comments which help a lot to improve our manuscript. We revised our manuscript according to the reviewer’s suggestion.

  1. It would be interesting if the authors discussed cell death by inflammation or pyroptosis, in which microglia may play an important role (paragraphs 220-226).

Our response: Again, we really appreciate the reviewer’s instructive comments. We revised our manuscript according to the reviewer’s suggestion.

  1. The article could increase its impact if Figures 2 and 3 are improved:
  • In Figure 2A, schematize the morphological differentiation of microglia, their surface markers, and the products secreted by each phenotype.
  • Figure 3 does not include insights or novel mechanisms through which microglia lead to synaptic dysfunction and memory impairment. The authors should improve this figure by adding the mechanisms described in the text: 1) microglial dysfunction in AD pathogenesis (genetic studies; TREM 2); 2) changes in microglial telomeres; 3) phagocytose Aβ through cell surface receptors; 4) aging-induced microglial dysfunction and its relationship with early synaptic loss (via C1q- and C3-mediated phagocytosis).
  • Incorporate in figure 3 and discuss in the text (could be included between lines 260 or 274) the possible activation of microglia by external stimuli that can mediate the activation of A1 or neurotoxic astrocytes and both lead to neuronal death (https://www.nature.com/articles/nature21029).
  • Make another figure or scheme adding the possible therapeutic targets on microglia activation (described in the manuscript) and their effect on synaptic dysfunction, for example: Inhibition of C1q, C3, and CR3; nonsteroidal drugs; minocycline, peroxisome proliferator-activated receptor-γ inhibitors; statins, folic acid, omega 3 fatty acids, cognitive training, physical exercise, nutrition intervention, etc. On the other hand, in the text discuss the adverse effects that could be triggered by inhibiting microglia and indirectly certain physiological roles.

Our response: Again, we really appreciate the reviewer’s detailed and instructive comments that help a lot to improve our manuscript. We modified all three figures and added more discussion according to the reviewer’s detailed suggestion.

  1. The authors mention that the role of the interaction between neurons and microglia (L185-186) is not clear. I suggest you review the following reference and discuss it: https://www.nature.com/articles/s41467-021-25486-7)

Our response: We updated the related information in revised manuscript according to the reviewer’s suggestion.

Minor concerns

  1. In the abstract, the phrase “new therapeutic” is repeated in the same sentence, change it to another.
  2. In keywords change AD for Alzheimer’s Disease
  3. In the introduction, highlight that AD is the most common neurodegenerative disorder worldwide
  4. Define Trem2, CD33, and ABCA7 (line 65)
  5. In L82 in the same sentence “different” is repeated

Our response: We revised our manuscript following the reviewer’s comments systematically.

Reviewer #2:

The authors discussed in detail about the relationship between Alzheimer’s disease and microglia. This review article clearly explained microglia's origin, function, mechanism related to AD, and clinical implications. As the clinical trials for drugs targeting Aβ failed, it is plausible to rethink the neuroinflammation or microglia dysfunction as sources of AD pathogenesis.

This paper provided plenty of details with good flow of writing. The content of the paper is a bit convoluted, but it is acceptable due to the complexity of the microglia’s involvement in AD.

I have some suggestions for this paper.

  1. Line 201, Interleukin-34 is not a neurotransmitter, although it is related to neural transmitter release. This should not be evidence of neurons affecting microglia by releasing neurotransmitters.

Our response: We totally agree with the reviewer that IL-34 is not a neurotransmitter. We revised our manuscript according to reviewer’s comments.

  1. Line 265 is the first time NSAID appears in the paper, please provide its full name.

 Our response: We revised our manuscript according to the reviewer’s suggestion.

  1. Line 283-284: “As this evidence…” Please make it clear what “this evidence” refers to.

Our response: We revised our manuscript according to the reviewer’s suggestion.

  1. Line 384-286, How the microglia involvement in Aβ relate to microglial dysfunction in this session. I can kind of get your reasoning, but please make it clear.

Our response: Thanks for the reviewer’s comment. We revised this part to make things more clear accordingly.

  1. Line 319-322, “Fourth … Aβ clearance”. These sentences are closely related to the previous paragraphs. You should better combine third and forth points, unless you have a clear argument that phagocytosis described in these two paragraphs are different.

Our response: We agree with the reviewer that these two paragraphs are closely related. We combined them together according to the reviewer’s suggestion.

  1.  Line 322-326, “Finally … pathology” these should be move to synaptic loss session afterwards.

Our response: We made changes according to the reviewer’s suggestion.

  1. Line 344, please remove “most”

Our response: We deleted it in revised manuscript.

  1. Line 360-360: “binding of …(Decker et al., 2010).” The reference (Decker et al., 2010) does not support the argument that binding of Aβ to postsynaptic glutamate receptors leads to synapse inactivation, and microglia cleave synapses by recognizing Aβ markers.

Our response: We apologize for the wrong citation. We added the right reference to support our argument during revision.

  1. Line 372-385: Most facts stated in this section is not microglia related, please reorganize this section or combine with the previous one.

Our response: We combined this section with the previous one according to the reviewer’s suggestion.

  1. Line 422: please provide full name of CSF1R

Our response: We provided the full name of CSF1R in the revised manuscript.

  1. Line 424-425: “Although how tau is engulfed by microglia is unclear,” No evidence presented shows Tau can be not engulfed by microglia.

Our response: We deleted this sentence in revised manuscript.

Reviewer 2 Report

The authors discussed in detail about the relationship between Alzheimer’s disease and microglia. This review article clearly explained microglia's origin, function, mechanism related to AD, and clinical implications. As the clinical trials for drugs targeting Aβ failed, it is plausible to rethink the neuroinflammation or microglia dysfunction as sources of AD pathogenesis.

This paper provided plenty of details with good flow of writing. The content of the paper is a bit convoluted, but it is acceptable due to the complexity of the microglia’s involvement in AD.

I have some suggestions for this paper.

1.       Line 201, Interleukin-34 is not a neurotransmitter, although it is related to neural transmitter release. This should not be evidence of neurons affecting microglia by releasing neurotransmitters.

2.       Line 265 is the first time NSAID appears in the paper, please provide its full name.

3.       Line 283-284: “As this evidence…” Please make it clear what “this evidence” refers to.

4.       Line 384-286, How the microglia involvement in Aβ relate to microglial dysfunction in this session. I can kind of get your reasoning, but please make it clear.

5.       Line 319-322, “Fourth … Aβ clearance”. These sentences are closely related to the previous paragraphs. You should better combine third and forth points, unless you have a clear argument that phagocytosis described in these two paragraphs are different.

6.       Line 322-326, “Finally … pathology” these should be move to synaptic loss session afterwards.

7.       Line 344, please remove “most”

8.       Line 360-360: “binding of …(Decker et al., 2010).” The reference (Decker et al., 2010) does not support the argument that binding of Aβ to postsynaptic glutamate receptors leads to synapse inactivation, and microglia cleave synapses by recognizing Aβ markers.

9.       Line 372-385: Most facts stated in this section is not microglia related, please reorganize this section or combine with the previous one.

10.   Line 422: please provide full name of CSF1R

11.   Line 424-425: “Although how tau is engulfed by microglia is unclear,” No evidence presented shows Tau can be not engulfed by microglia.

Author Response

(The authors gave the same response as above.)

Round 2

Reviewer 1 Report

Very good work done by the authors. It would only be necessary to add the respective abbreviations used in the figure captions.

Author Response

We do appreciate comments from the reviewer and the editor that help a lot to improve our manuscript. Again, we revised our manuscript accordingly. First, we added the respective abbreviations used in the figure captions. Second, we double-checked that all references were relevant to the contents of the manuscript. Any revisions made to the manuscript were marked up using the “Track Changes” function.